Characterization of the microbiome of the invasive Asian toad in Madagascar across the expansion range and comparison with a native co-occurring species

Santos Bárbara barbarasantosbio@gmail.com 1
Bletz Molly C. 2
Sabino-Pinto Joana 3
Cocca Walter 1
Fidy Jean Francois Solofoniaina 4
Freeman Karen LM 4
Kuenzel Sven 5
Ndriantsoa Serge 6
Noel Jean 4
Rakotonanahary Tsanta 6
Vences Miguel 3
Crottini Angelica 1
1 Cibio, Research Centre in Biodiversity and Genetic Resources, InBio, Universidade do Porto, Campus Agrário de Vairão, Rua Padre Armando Quintas, Portugal , Porto , Portugal
2 Department of Biology, University of Massachussetts Boston , Boston, MA , USA
3 Zoological Institute, Braunschweig University of Technology, Mendelssohnstr. 4, Germany , Braunschweig , Germany
4 Madagascar Fauna and Flora Group, BP 442, 501 Toamasina, Madagascar , Toamasina , Madagascar
5 Max Planck Institute for Evolutionary Biology, August-Thienemann-Str. 2, Germany , Plön , Germany
6 Amphibian Survival Alliance c/o Durrell Wildlife Conservation Trust, Madagascar Programme, Lot II Y 49 J 12 Ampasanimalo, BP 8511 101 Antananarivo, Madagascar , Antananarivo , Madagascar
Rollins Lee
Electronic publication date: 2021 Jun 28
Publication date: 2021
Volume: 9
Electronic Location ID: e11532
Received 2020 Oct 7; Accepted 2021 May 7
Copyright: ©2021 Santos et al.
Copyright year: 2021
Copyright holder: Santos et al.
License: This is an open access article distributed under the terms of the Creative Commons Attribution License, which permits unrestricted use, distribution, reproduction and adaptation in any medium and for any purpose provided that it is properly attributed. For attribution, the original author(s), title, publication source (PeerJ) and either DOI or URL of the article must be cited.
License URL: https://creativecommons.org/licenses/by/4.0/

Keywords: Duttaphrynus melanostictus, Ptychadena mascareniensis, Invasive species, Toamasina, Madagascar, 16s rRNA sequencing, Gut bacteria, Skin bacteria

Funding: The Portuguese National Funds through FCT (Foundation for Science and Technology) Investigador FCT (IF/00209/2014) 2020.00823. CEECIND PB/BD/106055/2015 SFRH/BD/102495/2014 National Funds through FCT/MCTES under the UIDB/50027/2020 Portuguese National Funds through FCT (Foundation for Science and Technology) supported the Investigador FCT (IF/00209/2014) and the 2020.00823. CEECIND contracts for Angelica Crottini and the doctoral fellowships for Bárbara Santos (PB/BD/106055/2015) and Walter Cocca (SFRH/BD/102495/2014). This work was also supported by National Funds through FCT/MCTES under the UIDB/50027/2020 funding. There was no additional external funding received for this study. The funders had no role in study design, data collection and analysis, decision to publish, or preparation of the manuscript.

==============================
Biological invasions are on the rise, with each invader carrying a plethora of associated microbes. These microbes play important, yet poorly understood, ecological roles that can include assisting the hosts in colonization and adaptation processes or as possible pathogens. Understanding how these communities differ in an invasion scenario may help to understand the host’s resilience and adaptability. The Asian common toad, Duttaphrynus melanostictus is an invasive amphibian, which has recently established in Madagascar and is expected to pose numerous threats to the native ecosystems. We characterized the skin and gut bacterial communities of D. melanostictus in Toamasina (Eastern Madagascar), and compared them to those of a co-occurring native frog species, Ptychadena mascareniensis, at three sites where the toad arrived in different years. Microbial composition did not vary among sites, showing that D. melanostictus keeps a stable community across its expansion but significant differences were observed between these two amphibians. Moreover, D. melanostictus had richer and more diverse communities and also harboured a high percentage of total unique taxa (skin: 80%; gut: 52%). These differences may reflect the combination of multiple host-associated factors including microhabitat selection, skin features and dietary preferences.

Introduction

Biological invasions can cause dramatic biodiversity loss (Enserink, 1999; Chornesky & Randall, 2003; Penk et al., 2016), with climate change, habitat alterations and direct anthropogenic translocation being the main factors facilitating the worldwide spread of alien, invasive species (Alpert, Bone & Holzapfel, 2000; Stachowicz et al., 2002; Walther et al., 2009; Crooks, Chang & Ruiz, 2011). Although alien invasive species do not always have detrimental effects (Schlaepfer, Sax & Olden, 2011), their potential devastating effects can be stronger in fragile island ecosystems. Here, they often out-compete or predate on local species, interfering with trophic networks and ultimately altering natural ecosystem function and balance (Lowe et al., 2000; Pitt, Vice & Pitzler, 2005). Among amphibians, notable invasive species include the Cane toad, Rhinella marina, the Puerto Rican Coquí, Eleutherodactylus coqui and the American bullfrog, Lithobates catesbeianus (Beard & Pitt, 2005; Shine, 2010; Snow & Witmer, 2010) of which the former is especially notorious for its negative effects in its invasive range in Australia (Shine, 2010). Among the many impacts that invasive species can have on native ecosystems, the introduction and spread of pathogenic fungi and viruses is also emerging as an important factor that may contribute to the global amphibian population decline (Miaud et al., 2016).

Microbiome research with high-throughput DNA sequencing techniques has enabled a better understanding of how host-associated microbiomes vary across host species, age, sex and habitats, and how their composition and diversity is influenced by host related and habitat-dependent factors (McKenzie et al., 2012; Bletz, Perl & Vences, 2017; Tiede et al., 2017). Symbiotic microbial communities likely inhabit all multicellular organisms and play an important role in the ecology, physiology, behaviour and health of their hosts (Dethlefsen, Mcfall-Ngai & Relman, 2007; Grice & Segre, 2012; Abdallah, Mijouin & Pichon, 2017; Lester et al., 2017). The skin microbiome can influence host’s ability to cope with environmental and habitat conditions and mediate immune responses (Sanford & Gallo, 2013; Grice, 2014; Jani & Briggs, 2018; Rebollar et al., 2016; Xavier et al., 2019), while gut-associated microbes can aid in food digestion, energy harvesting, development or immunity (Turnbaugh et al., 2006; Heijitx et al., 2011; Tuddenham & Sears, 2015). Microbiomes have been proposed to affect the host’s capacity for colonization, adaptation, and boosting the immune system (Rout et al., 2013; Gribben et al., 2017; Cheng et al., 2018). For example, interactions between invasive plants and associated microbes were found to suppress the rhizosphere microbes and other beneficial symbionts in native plants (Coats & Rumpho, 2014); on the contrary, some fungal symbionts were found to increase survival of their insect host, an invasive ant species, when these were exposed to pathogens (Konrad et al., 2014). In amphibians, only a few recent studies have evaluated the microbial communities in invasive species (e.g., Abarca et al., 2018; Christian et al., 2018; Kueneman et al., 2019).

Madagascar is one of the most celebrated biodiversity hotspots (Ganzhorn et al., 2001), known not only for the high degree of endemism but also for the ongoing habitat loss. Amphibian diversity in Madagascar is exceptionally high (Vieites et al., 2009; Perl et al., 2014; Brown et al., 2016) and severely threatened by habitat loss and human exploitation (Harper et al., 2007). Invasive species and pathogens in Madagascar are emerging as a new conservation concern since they may push native species further towards extinction (Bletz et al., 2015; Kull, Tassin & Carriere, 2015; Goodman et al., 2017). A naturalized population of the Asian common toad, Duttaphrynus melanostictus, was reported in Madagascar in 2014, and has since become a major conservation concern (Andreone et al., 2014; Crottini et al., 2014; Kolby, 2014). Originally from Asia, it is estimated to have been present in Madagascar since 2010, being first reported near the seaport city of Toamasina, on Madagascar’s eastern coast in 2014 (Fig. 1; (Kolby, 2014). Duttaphrynus melanostictus is believed to have arrived from Cambodia or Vietnam (Vences et al., 2017), possibly in shipping containers. At present, it occurs mainly in urban and rural landscapes with mixed Eucalyptus spp. forests, where native amphibian communities are highly impoverished. However, it is rapidly expanding (Licata et al., 2019; Licata et al., 2020) and it is feared that it may soon reach areas known to host richer amphibian communities such as Betampona Strict Nature Reserve and Parc Ivoloina (Rosa et al., 2012; Crottini et al., 2014).

Figure 1 Distribution of the sampling sites (Site 1 = green circle; Site 2 = orange triangle; Site 3 = blue square) visited in September 2016.

Known distribution area of the invasive population of Duttaphrynus melanostictus in Toamasina in late 2014 (yellow polygon; modified from Moore, Solofoniaina Fidy & Edmonds, 2015). The black square in the inset map shows the relative position of Toamasina in eastern Madagascar.

Although with low incidence, predation of smaller herpetofauna has been observed in other invasive populations of this toad (Döring et al., 2017), but the major concern of this invasion is associated with toad toxicity and the devastating effects that this species might have on its predators. In fact, D. melanostictus is known to produce highly poisonous skin toxins that are likely to negatively affect the vast majority of potential native predators (Marshall et al., 2018). Skin secretions isolated from D. melanostictus individuals collected in its native range seem to contain potent antimicrobial agents and important pharmacological compounds (Garg et al., 2007) that may increase disease resistance, making this invasive amphibian species a particularly interesting candidate for microbiome studies in invasive scenarios. Due to its life history traits that promote the capacity to reach high abundances and high dispersal rate, D. melanostictus is considered to have high invasive potential (Reilly et al., 2017; Licata et al., 2019; Licata et al., 2020). High invasive potential has been observed in other toad species such as R. marina, where this capacity has been linked to reduced ecological pressures in invaded areas and a rapid physiological adaptation to new environments (Phillips et al., 2006).

Here, we provide the first assessment of skin and gut bacterial communities of the invasive D. melanostictus population after its recent introduction to Madagascar and we compare its microbiome with that of a co-occurring native frog species, the Mascarene ridged frog (Ptychadena mascareniensis), across its expansion range. We hypothesized that the invasive species will present richer and more diverse bacterial assemblages due to intrinsic physiological and ecological characteristics, but also due to the recent introduction to a new environment (the invaded area). We also expect that across sites the toad may have different bacterial assemblages, due to the different arrival time in the new environment, while the native species, due to the limited geographic scale and similar habitat type, may exhibit similar bacterial assemblages. We explore the correlation between the toad’s microbial diversity and its high colonization and adaptation capacity, using as proxy measures of bacterial species richness, diversity and functional inference that may confer disease resistance or enhance wider diet range in different habitats.

Methods

Sampling

The study species collected were the invasive Asian toad (Duttaphrynus melanostictus) and the native and co-occurring species Mascarene ridged frog (Ptychadena mascareniensis). Sampling was performed in the invaded area around Toamasina (eastern Madagascar) (Fig. 1) between September 20th and 24th, 2016. All sampling sites are highly anthropogenically transformed areas and the selection of sampling sites was based on the known distribution of D. melanostictus at the time (retrieved from Moore, Solofoniaina Fidy & Edmonds (2015) and based on field observations carried out by the staff of Madagascar Fauna and Flora Group). We aimed to analyse sites where the toad established in different years (Fig. 1): Site 1 (green circle, S1), point where the toad was likely introduced around 2010; Site 2 (orange triangle, S2), site where the toad was not found in early 2014 and detected only in late 2014; Site 3 (blue square, S3), site that was recently colonized at the time of sampling (September 2016). In each site we collected 16 individuals (eight males and eight females) of D. melanostictus and four individuals of P. mascareniensis. Each specimen was collected with new nitrile gloves, measured (snout-vent length and weight), and kept in individual sterile plastic bags until sampling. Each specimen was rinsed with sterile water to remove debris and transient microbes, and swabbed 10 times on the ventral side and five times on each thigh and foot using one sterile swab (MW113, Medical Wire Equipment & Co. Ltd., Corsham, United Kingdom). Swabs were air dried, placed in their individual tube and kept at ca. 4 °C during the expedition and during the export from Madagascar, and transferred to −20 °C upon their arrival in Europe.

To characterize the gut bacterial communities for both species, at each site we used a solution of Tricaine Methanesulfonate (MS-222, Sigma-Aldrich) to euthanize 4 individuals of P. mascareniensis and four (two males and two females) of the 16 individuals of D. melanostictus. After euthanasia the specimens were dissected and the gut (the entire intestine portion including gut contents) was removed and stored in RNA later. For each site, we pooled the dissected guts of the four D. melanostictus into one tube and the dissected guts of P. mascareniensis into another tube. The six tubes with the pooled gut samples per site and per species were kept in liquid nitrogen during fieldwork, transferred to cool conditions (ca. 4 °C) during the export from Madagascar, and stored at −80 °C upon their arrival to Europe.

DNA extraction, amplification and sequencing

Gut tissue samples from each tube were homogenized. DNA from swabs and gut tissue was extracted following a modified Qiagen DNeasy Blood & Tissue Kit protocol (Hilden, Germany) with an initial lysozyme incubation step at 37 °C to break up cell walls of Gram-positive bacteria. To enable comparison of our data with previously published studies on the microbiome of Malagasy amphibians we amplified the V4 region of the bacterial 16S rRNA gene using the following primer set: 515F (5′-GTGCCAGCMGCCGCGGTAA-3′) and 806R (5′-GGACTACHVGGGTWTCTAAT-3′) (Caporaso et al., 2011). Amplification of each sample was performed in duplicate in a volume of 12.5 µl including 0.2 µl of Phusion Hot Start II DNA Polymerase (Thermo Fisher Scientific, Waltham, Ma, USA), 0.25 µl of each primer (10 µM), 0.25 µl of dNTPs, 2.5 µl of buffer, 8.1 µl of H2O and 1 µl of template DNA. The amplification protocol consisted of an initial denaturation step at 98 °C for 1 min, followed by 30 cycles of denaturation at 98 °C for 10 s, annealing at 55 °C for 30 s and elongation at 72 °C for 30 s, with a final extension at 72 °C for 5 min. The two PCR products of each sample were pooled together in a total volume of 25 µl and visualized on 1% agarose gel. All samples were pooled together according to band brightness and the final pooled sample was run in a 1% agarose gel and purified with QIAQuick Gel Extraction Kit (Qiagen, Hilden, Germany). Samples were sequenced using paired-end 2 x 250 v2 chemistry on an Illumina MiSeq sequencing platform using a dual-index approach (Kozich et al., 2013). Raw sequences were deposited in NCBI under the following BioProject ID PRJNA667830.

Sequence processing

Sequences were processed in Quantitative Insights into Microbial Ecology (QIIME v1.9.1) (Caporaso et al., 2010a). Due to the typical lower quality of reverse reads (Kwon et al., 2013), only the forward reads were filtered under the following criteria: absence of Ns within the sequence, absence of barcode errors, and exclusion of reads containing three or more consecutive low-quality nucleotides. Sequences were clustered into sub-operational taxonomic units (sOTUs, hereafter called OTUs) following the deblur workflow (https://github.com/biocore/deblur) (Amir et al., 2017). Sequences were trimmed to 150 bp and OTUs with less than 10 reads were excluded. The resulting OTUs were then assigned to a taxonomic group using the Greengenes 13.8 reference database (May 2013 release; https://greengenes.lbl.gov/). Non-bacterial taxa (e.g.: archaea, mitochondria and chloroplasts) were removed. All OTUs with less than 0.001% of the total reads of all analysed samples were excluded (Bokulich et al., 2013). PyNAST (Caporaso et al., 2010b) was used to align the OTU sequences and a phylogenetic tree was built with FastTree (Price, Dehal & Arkin, 2010). Data was organized into three datasets: Dataset A included only skin swabs from the two species; Dataset B included only skin swabs from males and females of D. melanostictus; and Dataset C included only gut samples from the two species (Table S1). Each dataset was rarefied to a specific number of reads per sample: Dataset A, B: 1,455; Dataset C: 1,867 (Table S1). Dataset A was additionally rarefied at 4,000 reads/sample to allow a better comparison with previous published works (e.g., Bletz et al., 2017; Kueneman et al., 2019). After filtering and rarefaction, the final Dataset A included a total of 37 samples (S1: 9 D. melanostictus, 4 P. mascareniensis; S2: 10 D. melanostictus, 3 P. mascareniensis; S3: 7 D. melanostictus, 4 P. mascareniensis) with 1,617 OTUs for the skin bacteria dataset. Dataset B included a total of 15 males (S1 =3, S2 =6, S3 =6) and 11 females (S1 =6, S2 =4, S3 =1) with the female samples from S3 being excluded from the analysis (Table S2). Dataset C included 3 pooled samples for each species (each containing the gut of 4 individuals per species per site) with 701 OTUs (Tables S1, S2).

Statistical analysis

Diversity indices and statistical analysis were performed using QIIME v1.9.1 and R v3.4.4 (R Core Team, 2016). Data was organized into three datasets. Dataset A included a total of 37 skin swabs from both species from all sites and was used to assess the effects of host and site on skin bacteria; Dataset B included 26 skin swabs from D. melanostictus from all sites and was used to assess the effect of sex on skin bacteria; Dataset C included six pooled (per species and per site) gut samples and was used to assess the effects of host species on the gut bacteria (Table S2 for more details).

Alpha diversity metrics were calculated to detect differences between host species, sexes and sites. Species richness was measured as number of observed OTUs (OTU Richness) and Chao1 diversity index; and diversity was measured using Shannon diversity index and Faith’s phylogenetic distance (PD). Significant differences between alpha indices were assessed using ANOVA (aov, stats package, R Core Team, 2016). For Dataset A, we used a two-way ANOVA, with factors “species”, “site” and their interaction; for Dataset B, we used a one-way ANOVA, with the factor “sex”. In Dataset C, we used a non-parametric Kruskal-Wallis Test (KW) using the variable “species” although the low sample size does not allow for a robust statistical analysis and values are indicative. Dissimilarity matrices were calculated using Weighted and Unweighted Unifrac distances (Lozupone & Knight, 2005) and visualized using a non-metric multidimensional scaling plot (NMDS, phyloseq package, McMurdie & Holmes, 2013). Differences in the bacterial community structure (Beta Diversity) were analysed with PERMANOVA (Adonis, vegan package, 999 permutations (Oksanen et al., 2016)) with species and site as predictor variables, including main effects and their interaction. When significant differences were observed, a test for homogeneity of groups dispersions was calculated using the function betadisper in vegan package in R (Oksanen et al., 2016). Community composition was visualized with bar plots including the most abundant taxa in each category (phylum, family and genus) after transforming the counts into relative abundances and grouping all other taxa with relative abundance lower than 1% (phylum from Dataset C), 5% (phylum from Datasets A and B), 15% (family and genus from Datasets A and B) and 10% (family and genus from Dataset C). An additional category “unidentified” represents the total relative abundance of taxa that were not identified at that taxonomic level. Total shared and unique OTUs for each species were represented as Venn diagrams for all groups using the collapsed biom tables retrieved from QIIME. Since no significant differences were found between sites, the subsequent analysis was performed with individuals from the three sites grouped together. Linear Discriminant Analysis Effect Size (LEfSe) method (LDA score > 3.0, α = 0.05) (Segata et al., 2011) was used to determine OTUs responsible for the observed differences in the skin and gut communities between species (Datasets A and C) and sexes (Dataset B). We used PICRUSt (Phylogenetic Investigation of Communities by Reconstruction of Unobserved States) (Langille et al., 2013) to gain a better understanding of the possible functions of the symbiotic bacteria identified in the skin and in the gut. The OTUs were assigned to the Greengenes v13.5 database using the 97% similarity with the closed OTU-picking strategy, and a normalization of the copy numbers of each OTU was performed. Subsequently, the metagenome of each sample was predicted, and a functional categorization with respective abundances (following the Kyoto Encyclopedia of Genes and Genomes –KEGG –Orthology database) performed, using level 2 KEGG Orthologs (KO). Pathways with less than 10 counts were removed and abundances were rarefied. Both LEfSe and PICRUSt analysis were run on the Galaxy Web platform (http://huttenhower.sph.harvard.edu/galaxy). Significant differences between host species were assessed using the Kruskal-Wallis test (K-W) in QIIME. To better understand if the bacterial taxa could provide advantages regarding higher disease resistance in the toad, we mapped all skin bacterial OTUs (Datasets A and B) using a closed-reference OTU picking strategy, against the published database of antifungal amphibian skin bacterial isolates (Woodhams et al., 2015) - this database includes isolates that are likely able to inhibit or enhance the growth of the amphibian fungal pathogen, Batrachochytrium dendrobatidis (Bd). Taxa with a match of 97% were retrieved and the proportions of OTUs with putative Bd-inhibitory or Bd-enhancing properties were calculated.

All applicable international, national and/or institutional guidelines for the care and use of animals were followed. Ministère de l’Environnement et du Développement Durable provided the research permits for: collection, N ∘226/16/MEEF/SG/DGF/DSAP/SCB.Re of September 19th, 2016; transport, N°1679-16/MEEF/SG/DGF/DREEF.ATS/SREco and N°1680-16/MEEF/SG/DGF/DREEF.ATS/SREco of September 24th, 2016; and export, N°284N-EA10/MG16 of October 5th.

Results

Dataset A - Comparison of the skin microbiome of Duttaphrynusmelanostictus and Ptychadena mascareniensis across the expansion range

Host species had a significant effect on alpha diversity indices of the cutaneous microbiome (Figs. 2A–2D): D. melanostictus showed significantly higher values for OTU richness (ANOVA, F = 33.15, p < 0.001), phylogenetic diversity (ANOVA, F = 40.66, p < 0.001), Chao1 diversity (ANOVA, F = 29.64, p < 0.001) and Shannon diversity (ANOVA, F = 7.289, p = 0.006) compared to the native P. mascareniensis. Site did not have an effect on alpha diversity (ANOVA, OTUs: F = 0.505, p = 0.61; PD: F = 1.830, p = 0.180; Chao1: F = 1.274, p = 0.30; Shannon: F = 1.074, p = 0.36, SM Fig. S1A), and the interaction between species and site was not statistically significant (ANOVA, OTUs: F = 0.729, p = 0.49; PD: F = 1.078, p = 0.35; Chao1: F = 0.334, p = 0.72; Shannon: F = 0.783, p = 0.47). However, a trend was observed with D. melanostictus showing greater values in all alpha indices across sites while P. mascareniensis showed an irregular pattern.

Figure 2 Skin bacterial diversity and composition of Duttaphrynus melanostictus and Ptychadena mascareniensis across sites.

Alpha diversity: (A) OTU Richness. (B) Shannon index. (C) Chao1 diversity. (D) Phylogenetic diversity. Different letters (a, b) indicate significant different groups; Beta diversity: (E) Non-metric multidimensional scaling (NMDS) ordination of Weighted Unifrac Distances, (F) Non-metric multidimensional scaling (NMDS) ordination of Unweighted Unifrac Distances; Abundance plots: Composition of the skin bacterial communities including the most abundant taxa from each taxonomic level: phylum (top), family (middle) and genus(bottom) in (G) D. melanostictus and (H) P. mascareniensis across the three sites. Photo credit: Angelica Crottini, Javier Lobon-Rovira.

Beta diversity significantly differed between host species when measured by both weighted Unifrac (Fig. 2E, PERMANOVA: Pseudo- F(1,36) = 4.896, R2 = 0.118, p = 0.002) and unweighted Unifrac metrics (Fig. 2F, PERMANOVA: Pseudo- F(1,36) = 6.565, R2 = 0.156, p = 0.001); but did not differ across sites (Figs. 2E–2F, PERMANOVA: weighted Pseudo- F(2,36) = 1.138, R2 = 0.055, p = 0.32; unweighted Pseudo- F(2,36) = 1.036, R2 = 0.049, p = 0.36). Similarly, the interaction of species and site did not affect beta diversity (Figs. 2E–2F, PERMANOVA: weighted Pseudo- F(2,36) = 1.664, R2 = 0.080, p = 0.06; unweighted Pseudo- F(2,36) = 1.217, R2 = 0.058, p = 0.15). Analysis of dispersion indicated no significant differences between species (Weighted: F(1,35) = 0.863, p = 0.37; Unweighted: F(1,35) = 1.996, p = 0.17) or sites (Weighted: F(2,34) = 0.0649, p = 0.94; Unweighted: F(2,34) = 0.943, p = 0.40).

The skin bacterial communities from the two species were mainly composed of the same phyla (Actinobacteria, Bacteroidetes and Proteobacteria) but with several differences in relative abundances at lower taxonomic levels (family and genus; Figs. 2G–2H). The D. melanostictus skin community had higher abundances of the families Alteromonadaceae, Comamonadaceae, Moraxellaceae and Sphingobacteriacae while the P. mascareniensis skin community had higher abundances of Enterobacteriaceae, Moraxellacaeae (only at Site 3), Pseudomonadaceae and Xanthomonadaceae. Notably, P. mascareniensis had a higher abundance of bacteria of the genus Pseudomonas while D. melanostictus had Cellvibrio as the most abundant genus. Across sites, the differences observed between host species were concordant. Within species, the patterns varied: D. melanostictus skin bacterial communities were more stable across sites and P. mascareniensis showed more variability in taxonomic abundance (Figs. 2G–2H). LEfSe analysis revealed 39 taxa that were differently abundant in the two host species including 13 taxa that exhibited higher relative abundance in P. mascareniensis and 26 in D. melanosticus (Fig. 3). Specifically, P. mascareniensis only had differently abundant taxa from the phylum Proteobacteria and only one from the phylum Firmicutes, while D. melanostictus was characterized by significant differential abundance of taxa from the Actinobacteria, Proteobacteria, Bacteroidetes and Verrucomicrobia phyla. In P. mascarenienis, all differentially abundant taxa were included within the class Gammaproteobacteria with the exception of one Alphaproteobacteria taxon. In the case of D. melanostictus, differentially abundant bacteria belonged to several classes and families within different phyla (Fig. 3A).

Figure 3 Differently abundant skin taxa occurring in Duttaphrynus melanostictus (yellow bars) and Ptychadena mascareniensis (blue bars).

–LDA scores of detected OTUs in LEfSe analysis. Photo credit: Angelica Crottini, Javier Lobon-Rovira.

In total, D. melanostictus had more than 1,000 unique OTUs (equivalent to 80% of total number of OTUs) and shared only 238 (15%) with P. mascareniensis, while the latter had only 5% unique OTUs (Fig. S1A). The percentage of shared OTUs between the species was similar at sites 1 and 3 and lower at site 2 (Fig. S1C). This lower percentage was coupled with higher number of unique OTUs found in the toad (Fig. S1C). Individuals of D. melanostictus across sites shared between 30–40% of OTUs, while the percentage of unique OTUs found at each site was around 20% (Fig. S1B); individuals from sites 1 and 2 shared more OTUs than in comparison with the number of shared OTUs between each of the first two sites with site 3 (Fig. S1B). P. mascareniensis had similar trends, with individuals sharing a higher number of OTUs compared with the unique OTUs found at each site. However, the percentage of OTUs found at all three sites was only 9% (Fig. S1B).

A total of 39 KEGG pathways (Level 2) were predicted for the two amphibians’ skin microbiomes, of which 18 exhibited significantly different relative abundance between species (Table S3). From these, D. melanostictus had 11 enriched functional groups including cell growth and death, transport and catabolism, biosynthesis of secondary metabolites, energy and lipid metabolism, xenobiotics biodegradation and environmental adaptation. P. mascareniensis had seven enriched functional groups including membrane transport, infectious diseases, cellular processes and signalling, among others.

The two amphibians had significant differences in the proportion of OTUs with putatively Bd-inhibitory capacities but not of Bd-enhancing skin OTUs (K-W: χ2 = 11.5, p < 0.001 and χ2 = 3.10, p = 0.078 respectively) with P. mascareniensis carrying higher proportions of putative Bd-inhibitory OTUs (Fig. S3A).

Dataset B - Comparison of the skin bacterial community of males and females of Duttaphrynus melanostictus

In terms of alpha diversity, Shannon was the only metric that was significantly different between the sexes (ANOVA: p = 0.04, Figs. 4A–4D). Beta diversity showed that sex was significant when assessing weighted Unifrac distances (PERMANOVA: Sex, Pseudo- F= 2.35, R2 = 0.09, p = 0.02, Fig. 4E) but not with the Unweighted Unifrac distances (PERMANOVA: Sex, p > 0.05, Fig. 4F). Analysis of dispersion indicated a significant difference between sexes (Weighted Unifrac, F(1,24) = 4.8089, p = 0.032).

Figure 4 Skin bacterial diversity and composition of males (green) and females (orange) of Duttaphrynus melanostictus.

Alpha diversity: (A) OTU Richness. (B) Shannon index. (C) Chao1 diversity. (D) Phylogenetic diversity. Different letters (a, b) indicate significant different groups; Beta diversity: (E) Non-metric multidimensional scaling (NMDS) ordination of Weighted Unifrac Distances. (F) Non-metric multidimensional scaling (NMDS) ordination of Unweighted Unifrac Distances; Abundance plots: Composition of the skin bacterial communities including the most abundant taxa from each taxonomic level phylum (top), family (middle) and genus (bottom) in (G) females and (H) males. Photo credit: Angelica Crottini.

Males had higher abundances of Sphingobacteriaceae and a high rate of unidentified taxa when compared with females (Figs. 4G–4H). At the genus level, sex seemed to influence the abundance level of the most common taxa (Arthrobacter, Cellvibrio, Devosia) but without a clear pattern. Once again, males had a higher abundance of unidentified genus than females (Figs. 4G–4H). However, LefSe analysis indicated that there were no differently abundant taxa.

Males had double the number of unique OTUs compared to females when samples from the three sites were grouped (Fig. S2). A total of five predicted KEGG pathways (Level 2) were more abundant in males and four in females (Table S4). Among these, females exhibited significantly higher abundances of functional groups associated with Immune System Diseases while males had higher abundances of functional groups associated to other diseases and associated to cellular processes (e.g., Transport and Catabolism) (Table S4).

Comparing the skin bacterial communities with the antifungal database showed marginal differences with males exhibiting a slightly higher proportion of putative Bd-inhibitory skin OTUs (K-W: χ2 = 3.59, p = 0.058). Both sexes exhibited similar proportions of putative Bd-enhancing skin OTUs (K-W: χ2 = 0.12, p = 0.74) (Fig. S3B).

Dataset C - Comparison of gut bacterial communities of Duttaphrynus melanostictus and Ptychadena mascareniensis

In dataset C, only one pooled sample (with four individuals each) per site and species was obtained, thus all the statistics were performed to compare only the effect of the host species (Figs. 5A–5D). The complete plots with separated sites are available in supplementary material (Figs. S4A–S4D). The gut communities did not present significant differences in alpha diversity between host species for any of the indices (KW, OTUs: χ2 = 2.33, df = 1, p = 0.13; Chao1: χ2 = 1.19, df = 1, p = 0.28; PD: χ2 = 1.19, df = 1, p = 0.28; Shannon: χ2 = 1.19, df = 1, p = 0.28) although a trend for an increase in bacterial richness and diversity was observed in D. melanostictus (Figs. 5A–5D; SM Fig. S4A).

Figure 5 Gut bacterial diversity and composition of Duttaphrynus melanostictus (yellow) and Ptychadena mascareniensis (blue).

Alpha diversity: (A) OTU Richness. (B) Shannon index. (C) Chao1 diversity. (D) Phylogenetic diversity. No significant differences; Beta diversity: (E) Non-metric multidimensional scaling (NMDS) ordination of Weighted Unifrac Distances, (F) Non-metric multidimensional scaling (NMDS) ordination of Unweighted Unifrac Distances; Abundance plots: Composition of the skin bacterial communities including the most abundant taxa from each taxonomic level phylum (top), family (middle) and genus (bottom) in (G) D. melanostictus and (H) P. mascareniensis. Photo credit: Angelica Crottini, Javier Lobon-Rovira.

No significant differences in gut community composition were found between the two species using both weighted Unifrac (PERMANOVA: Pseudo- F1,5 = 1.66; R2 = 0.3, p = 0.30) and unweighted Unifrac (PERMANOVA: Pseudo- F1,5 = 1.75; R2 = 0.3, p = 0.10) distances (Figs. 5E–5F, Fig. S4B).

The gut community of both species was dominated by three phyla (Bacteroidetes, Proteobacteria and Firmicutes) (Figs. 5G–5H; Fig. S4C), with differences in the relative abundances at lower taxonomic levels. D. melanostictus also exhibited high relative abundance of Fusobacteria. At the family level, D. melanostictus showed a more diverse gut community in terms of relative abundance including 10 families almost equally abundant (Figs. 5G–5H), while P. mascareniensis had 7 families of high relative abundances, and among these, Clostridiaceae and Streptococcaceae were the most abundant. At the genus level, the gut community of D. melanostictus exhibited high abundance of Bacteroidetes and Cetobacterium while P. mascareniensis gut community was dominated by higher abundances of Clostridium and Lactococcus. Across sites, the gut communities of both amphibian species also exhibited significant differences in terms of relative abundances of several taxa (Fig. S4C). With LefSe analysis, a total of 22 taxa were identified as being significantly more abundant in the gut of D. melanostictus, including members of the three phyla (Bacteroidetes, Firmicutes, Proteobacteria) while no taxa were enriched in P. mascareniensis (Fig. 6).

Figure 6 Differently abundant gut taxa occurring in Duttaphrynus melanostictus with LDA score.

No differently abundant gut taxa were identified in Ptychadena mascareniensis. Photo credit: Angelica Crottini.

D. melanostictus had 365 unique OTUs (52%), while P. mascareniensis had 256 (37%) and only 11% of the bacterial OTUs (corresponding to a total of 80 OTUs) was shared (Fig. S5).

From the gut communities of the two species a total of 39 KEGG pathways (Level 2) were predicted, with both amphibians exhibiting the same functional groups. Significant differences in the abundance levels of these pathways were detected (K-W, p-value < 0.05) (Table S5). D. melanostictus had four enriched functional groups: Biosynthesis of Secondary Metabolites, Energy Metabolism, Endocrine System and Information Processing –Folding, Sorting and Degradation, while P. mascareniensis had only one enriched functional group associated with membrane transport (Table S5).

Discussion

Our study provides the first characterization of the skin and gut microbiomes of the Asian common toad Duttaphrynus melanostictus in its invasive range in Madagascar occurring in a highly human impacted area, and includes a comparison with the co-occurring native species Ptychadena mascareniensis. To our knowledge, only five recent studies characterized the microbiome of an invasive amphibian: Christian et al. (2018) found that Rhinella marina had the poorest and most dissimilar skin bacterial community in comparison with native amphibians in Australia; Abarca et al. (2018) found, for the same species, higher skin bacterial diversity in individuals from the invaded range compared to the native range; while, Kueneman et al. (2019) found that on a global scale the skin microbiome of Lithobates catesbeianus was more similar to that of the native amphibians than to itself in different parts of its invasive range. Two more recent studies focused on the gut microbiome of invasive species. The first one characterized the microbiome of R. marina while comparing the bacterial community across gut sections and found that sex influenced the gut microbiota and that cloacal swabs can be a good proxy to study intestinal microbes (Zhou et al., 2020). The second study found that the gut microbiome of the invasive guttural toad (Sclerophrys gutturalis) exhibited greater microbial diversity and functional flexibility when compared with bacterial communities from the native populations (Wagener, Mohanty & Measey, 2020). We investigated a very recent invasion and aimed at characterizing the microbiome of D. melanostictus across its expansion range and how it differs from the microbiome of a native species from across three sites with similar levels of human impact. Skin bacterial communities were strongly correlated with host species, with D. melanostictus showing higher richness and diversity. To a lesser extent, sex also influenced these communities. However, no significant differences in the skin bacterial composition were observed between sampling sites for both species, which might be related to the small geographic area and the overall similar habitat of the three sampling sites (all sampling sites were urban areas). It also indicates that the skin microbiome of D. melanostictus individuals has likely been stable across its early expansion.

Skin bacteria diversity differs between the invasive toad D. melanostictus and the native frog P. mascareniensis but not across sites

The skin bacterial community of the two amphibian species differed in terms of richness, diversity, community structure and functional inferences. D. melanostictus hosted a richer community than the co-occurring native species, harboured many unique OTUs (80%) and average richness values similar to those found for other terrestrial species of amphibians in Madagascar (Table S6; Bletz et al., 2017; Kueneman et al., 2019). On the other hand, the native species P. mascareniensis showed lower values of bacterial richness than the average values found in previous studies comparing terrestrial or aquatic species from Madagascar (Table S6; Bletz et al., 2017; Kueneman et al., 2019). However, previous studies mostly included terrestrial amphibians (including P. mascarenienis) from multiple habitats (Bletz et al., 2017), while our study included only urban sites around Toamasina. This habitat is characterized by high anthropogenic pressures, such as the presence of cattle, human waste, no natural vegetation cover and poor availability of clean water bodies, that can potentially have impoverished the environmental bacterial pool and consequently reduced the richness of the bacterial communities in P. mascareniensis (Becker et al., 2017; Jiménez et al., 2020) but this should be further investigated in future assessments.

Notably, D. melanostictus showed a dramatically higher percentage of unique OTUs (80%) compared to the native species, which may be related with the toad skin characteristics and therefore different skin microenvironment, and other host ecological factors. The percentage of shared bacterial taxa (15%) between our two species is low compared with that observed in previous works comparing different species (25–70%), or between aquatic and terrestrial ecomorphs or different life stages (Rebollar et al., 2016; Bletz et al., 2017; Kueneman et al., 2014). Although not significant, we observed some variation in alpha diversity levels in the skin microbiota of D. melanostictus in the three analysed sites. This could be linked to different environmental bacteria colonizing the skin in each site. This was accompanied by higher percentage of shared OTUs with P. mascareniensis which can also support the hypothesis that the toad is being colonized by new environmental bacteria. To explain differences in bacterial composition between the two target species, skin texture may play a major role. Tubercles in D. melanostictus skin may provide alternative microniches for the bacteria compared to the smoother skin of P. mascareniensis. Moreover, the skin of amphibians has been suggested to select and filter for specific bacteria from the surrounding environment due to the secretion of skin compounds that may block colonization by some taxa and favour others, and this selection could differ among host species (Flechas et al., 2019; Walke et al., 2014). Terrestrial amphibian species (such as D. melanostictus) are expected to have richer skin communities than aquatic or arboreal amphibians (Bletz et al., 2017; De Assis, Barreto & Navas, 2017; Kueneman et al., 2019; Walke et al., 2014) partly because the soil usually harbours a richer bacterial pool than aquatic systems and the habitat is known to greatly influence amphibians’ skin communities. Although both species were found in the highly anthropized areas in Toamasina, P. mascareniensis were mostly found in the grass and often close to small water-bodies, while individuals of D. melanostictus were conspicuous within villages, sometimes near domestic animals or anthropogenic waste.

So far, only one recent study attempted to assess the role of bacterial communities in the adaptation of amphibians to novel habitats by studying the gut communities in invasive and native populations (Wagener, Mohanty & Measey, 2020). In other systems, more diverse microbiomes have been linked to higher host fitness, such as, for instance, pathogen resistance in wheat (Matos, Kerkhof & Garland, 2005), or defence against chemical compounds in beetles (Cheng et al., 2018). In amphibians, richer microbiomes have been linked to a higher resistance to pathogens (Becker & Harris, 2010; Harrison et al., 2017) and a richer microbiome could conceivably aid in the colonization of novel habitats (Wagener, Mohanty & Measey, 2020). Bacterial taxa associated with disease resistance were among the most abundant groups in both hosts but with specific differences in taxa identity and abundances. P. mascareniensis seems to carry a more diverse bacterial community with antifungal properties while the toad carried more OTUs from the Comamonadaceae family that contains taxa used in probiotic assays (Becker et al., 2015). D. melanostictus also had very low abundance of Pseudomonas, a genus that is ubiquitous in the environment (soil, water), plants and other organisms and is linked to resistance to pathogens like Bd (Becker et al., 2015). Pseudomonas was highly prevalent in P. mascareniensis and is usually abundant in amphibians from tropical regions (Bletz et al., 2017). Notably, the low abundance of Pseudomonas found in D. melanostictus agrees with the pattern found in invasive populations of R. marina: lower in invasive populations compared with native ones (Abarca et al., 2018), and lower in comparison with co-occurring native amphibians from Australia (Christian et al., 2018) and further analysis of this similarity should be applied.

The functional redundancy here observed was congruent with previous studies (Bletz et al., 2016; Huang et al., 2018), demonstrating that different microbiome assemblages from different hosts can succeed in the same environment and are probably more associated with host identity. From the host’s perspective this is crucial since it means that it maintains functional stable microbial community despite carrying different bacterial assemblages. The functional category of xenobiotics biodegradation and metabolism that was enriched in D. melanostictus could be related with a high capacity to cope with environmental alteration and anthropogenic stress which would be the case in Toamasina, and therefore higher adaptability or resilience to highly impacted or new habitats (Claus, Guillou & Ellero-Simatos, 2016). A future comparison with individuals occurring in less human impacted areas could provide more insights regarding the functional capacity of the microbiome in relation to the host habitat.

In a recent study with R. marina collected near its invasion front and where chytridiomycosis is absent, it was observed that individuals had lower Bd-inhibitory bacteria when compared with areas where Bd was present, highlighting the hypothesis that these bacteria are selected when the pathogen is present (Weitzman et al., 2019). In the species analysed here, the lower proportion of bacteria with putative Bd-inhibiting functions in D. melanostictus (in comparison with P. mascareniensis) may be related with its occurrence is Toamasina where Bd has not yet been detected (Bletz et al., 2015). However, it is important to note that the antifungal database include taxa isolated from amphibian species from Africa, America and Australia with no representatives from Asia, which can partially be responsible for the high proportion of taxa found in the native P. mascareniensis (Woodhams et al., 2015). Moreover, this classification is based on 97% similarity to bacteria that can inhibit Bd which does not necessarily mean that these bacteria actually have this function.

Moreover, D. melanostictus has been showed to have a high Bd prevalence (43%) in its native areas in India (Thorpe et al., 2018), which may be linked to a low prevalence of Bd-inhibitory bacteria, although it has not been tested there. A screen of the microbiome diversity of D. melanostictus from its native areas, and where Bd has been detected, could give new insights about microbiome patterns.

Gut bacteria show no differentiation between the two species

In gut bacterial communities some patterns were similar to skin communities. For example, D. melanostictus individuals hosted a bacterial gut community characterized by more unique OTUs and higher richness values (although not statistically significant). Although both species are generalist feeders (Döring et al., 2017; Fatroandrianjafinonjasolomiovazo et al., 2011), P. mascareniensis feeds mainly on arthropods while the diet of D. melanostictus includes other invertebrates and occasionally also small vertebrates (e.g., worm snakes) (Hahn, 1976; O’Shea et al., 2013). A larger body size probably allows the consumption of larger and more diverse prey whereas the microhabitat type (soil, water, leaves) may also hold different invertebrate groups influencing the potential prey availability for the two species. A richer bacterial community has been related with richer diets probably aiding the host in the digestion and metabolization of different items (Tiede et al., 2017; Wagener, Mohanty & Measey, 2020). The dominant bacterial phyla identified in the guts of the two amphibian species were similar to other studies (Fig. 5C) (Chang et al., 2016; Huang et al., 2018), which might be explained by the stable gut environment across species (compared for instance to the external environment). The relative abundance of taxa, however, varied between the two species probably associated with gut physiology, host diet and habitat conditions although the low sample size and the fact that the samples were pooled prevents us from obtaining robust comparisons (Ley et al., 2008; Tiede et al., 2017; Zhang et al., 2010). Members of the phyla Firmicutes (mainly belonging to the class Clostridia) are linked to fermentation of carbohydrates and found to be common in terrestrial animals, thus its high occurrence in P. mascareniensis was expected. D. melanostictus has a longer gut, and the lower oxygen availability associated with this environment (in addition to the host’s generalist diet), might explain the dominance of Bacteroidetes (Döring et al., 2017; Nelson et al., 2013). Bacteroidetes can also assist in metabolizing different energy sources (Flint et al., 2012). The higher proportion of members of the family Desulfovibrionaceae in D. melanostictus should be further studied since the group includes taxa that can be opportunist pathogens and produce endotoxins (Zhang et al., 2010). The overall absence of significant differences between the gut communities of these two species could also be the result of the low sample size used for this dataset, even if each pool included a mix of gut samples from 4 individuals. Besides the lack of significance in alpha and beta diversity, it is worth noting that we observed differences in the relative abundance levels, number of unique OTUs and functional inference, but further investigation is needed in this respect.

Conclusions

The expansion of the Asian common toad Duttaphrynus melanostictus in Madagascar is ongoing and comparing the recently introduced populations across its invaded range and subsequent expansion into different habitat types could help understand how microbiome changes through the process of invasion in a contemporary scenario. Our study shows that the skin microbiome of D. melanostictus is richer and more diverse than the skin microbiome of the native species, and this diversity is probably associated with the toad’s intrinsic physiological and ecological traits (e.g., the toad’s skin microenvironment), or can be linked to the colonization of new areas, with the toad being less likely to be selecting for any specific taxa. Expanding this study to other native amphibian species and other habitats (as shown in Licata et al., 2020, the toad is currently occupying urban, rural/agricultural, palm-oil plantation and savoka (degraded forest and mixed scrubland) habitats) is needed to further understand these differences. Additional data from its natural range in Asia and from other invasive populations could help to better characterize the degree of variation between native and invasive populations. For Madagascar, we encourage the development of new studies aimed at characterizing skin secretions and the antifungal properties of the skin microbiome of D. melanostictus. Similarly, we think that it will be beneficial to further investigate the connection between the toad’s diet and its gut microbiome (composition and functional roles), especially in this invasion scenario where the host may have to adapt fast as it will expand to different areas in Madagascar (colonization of different habitats).

Supplemental Information

Supplemental Information 1 Skin bacterial diversity of Duttaphrynus melanostictus and Ptychadena mascareniensis

(A) Total shared and unique OTUs between the two species. (B) Total shared and unique OTUs within each species across sites. (C) Total shared and unique OTUs between species in each site. Photo credit: Angelica Crottini, Javier Lobon-Rovira.

Click here for additional data file.

Supplemental Information 2 Total of shared and unique OTUs in males and females of Duttaphrynus melanostictus (individuals from the three sites were grouped)

Click here for additional data file.

Supplemental Information 3 Proportion of OTUs with putative Bd-Inhibitory and Bd-Enhancing microbiota compared to the Antifungal Isolates Database (Woodhams et al., 2015)

(A) Proportions of Bd-Inhibitory (yellow) and Bd-Enhancing (purple) skin OTUs from Duttaphrynus melanostictus and Ptychadena mascareniensis; (B) Proportions of Bd-Inhibitory (yellow) and Bd-Enhancing (purple) skin OTUs from females and males of Duttaphrynus melanostictus. Asterisks denote significant differences; “ns” denote non-significant differences Photo credit: Angelica Crottini, Javier Lobon-Rovira.

Click here for additional data file.

Supplemental Information 4 Gut bacterial diversity of Duttaphrynus melanostictus and Ptychadena mascareniensis

A-D) Alpha Diversity metrics were all significantly different between host species (p < 0.05) but not across sites (p > 0.05); E–F) Skin bacterial community structure of D. melanostictus (circles) and P. mascareniensis (triangles) across the 3 sites using a Non-metric multidimensional scaling (NMDS) ordination of Weighted and Unweighted Unifrac Distances; G–H) Composition of the skin bacterial communities including the most abundant Phyla (top), families (middle) and genera (bottom) in D. melanostictus (left panel) and P. mascareniensis (right panel) across the 3 sites. Photo credit: Angelica Crottini, Javier Lobón-Rovira.

Click here for additional data file.

Supplemental Information 5 Total shared and unique OTUs in the gut communities of Duttaphrynus melanostictus and Ptychadena mascareniensis. Samples from the three sampling sites were pooled together for each species

Click here for additional data file.

Supplemental Information 6 Total number of samples, sequences and OTUs available in each dataset before and after each filter

Click here for additional data file.

Supplemental Information 7 Samples available in datasets A, B and C after sequence processing. For dataset A, values correspond to the number of samples obtained after rarefaction (1,455 or 4,000).

Click here for additional data file.

Supplemental Information 8 Predicted abundance of KEGG ortholog groups (Level 2 KOs) from skin bacterial communities of Duttaphrynus melanostictus and Ptychadena mascareniensis

Groups that present significant difference (Kruskal-Wallis test) in the abundance levels between host species are colored. In yellow are the groups that were more abundant in D. melanostictus and in blue the groups more abundant in P. mascareniensis.

Click here for additional data file.

Supplemental Information 9 Predicted abundance of KEGG ortholog groups (Level 2 KOs) from skin bacterial communities of males and females of Duttaphrynus melanostictus

Groups that show significant higher abundance levels (Kruskal-Wallis test) in males (green) and females (orange) are highlighted.

Click here for additional data file.

Supplemental Information 10 Predicted abundance of KEGG ortholog groups (Level 2 KOs) from gut bacterial communities of Duttaphrynus melanostictus and Ptychadena mascareniensis that present significant difference (Kruskal-Wallis test) in the abundance levels between host

Colored in yellow are the groups that were more abundant in D. melanostictus and in blue groups more abundant in P. mascareniensis.

Click here for additional data file.

Supplemental Information 11 Supplemental_Table_S6.docx SUPPORTING TABLE (23KB) Comparison of average values of bacterial richness (number of OTUs) obtained in this this study with two previous recent studies on Malagasy terrestrial and aquatic amphibians

In accordance with previous studies only results of dataset A (after rarefaction at 4,000 reads) are shown.

Click here for additional data file.

Additional Information and Declarations

Competing Interests

Author Contributions

Animal Ethics

Field Study Permissions

Data Availability

The authors declare there are no competing interests.

Bárbara Santos conceived and designed the experiments, performed the experiments, analyzed the data, prepared figures and/or tables, authored or reviewed drafts of the paper, and approved the final draft.

Molly Bletz and Joana Sabino-Pinto analyzed the data, prepared figures and/or tables, authored or reviewed drafts of the paper, and approved the final draft.

Walter Cocca, Jean Francois Solofoniaina Fidy, Serge Ndriantsoa, Jean Noel and Tsanta Rakotonanahary performed the experiments, authored or reviewed drafts of the paper, and approved the final draft.

Karen L.M. Freeman conceived and designed the experiments, authored or reviewed drafts of the paper, and approved the final draft.

Sven Kuenzel and Miguel Vences analyzed the data, authored or reviewed drafts of the paper, and approved the final draft.

Angelica Crottini conceived and designed the experiments, performed the experiments, authored or reviewed drafts of the paper, and approved the final draft.

The following information was supplied relating to ethical approvals (i.e., approving body and any reference numbers):

Ministère de l’Environnement et du Développement Durable approved all the research permits:

Collection: No 226/16/MEEF/SG/DGF/DSAP/SCB.Re of September 19th, 2016

Transport: No 1679-16/MEEF/SG/DGF/DREEF.ATS/SREco and No 1680-16/MEEF/SG/DGF/DREEF.ATS/SREco of September 24th, 2016

Export: No 284N-EA10/MG16 of October 5th.

The following information was supplied relating to field study approvals (i.e., approving body and any reference numbers):

Ministère de l’Environnement et du Développement Durable provided the research permit (N° 226/16/MEEF/SG/DGF/DSAP/SCB.Re of September 19th, 2016).

The following information was supplied regarding data availability:

Data are available at NCBI: BioProject ID PRJNA667830.

The raw sequences and metadata are available at Zenodo: Bárbara Santos. (2021). PeerJ49550. http://doi.org/10.5281/zenodo.4518953.

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
