# Peer review of "Characterization of the microbiome of the invasive Asian toad in Madagascar across the expansion range and comparison with a native co-occurring species"

_PeerJ, doi:10.7717/peerj.11532_

## Round 0.1 · original submission · Major Revisions

Thank you for this interesting submission. The reviewers have made a substantial number of useful comments to improve your manuscript. Please carefully consider and address each of these comments prior to resubmission. In particular, please attend to the comments regarding pseudoreplication. Further, I agree that streamlining your manuscript will enable your future readers to better understand your message.

In addition, we have recently published a Rhinella marina gut microbiome paper that was not on your list of invasive amphibian microbiome studies and may be of some use to you with respect to interpreting your data (Zhou et al. 2020, Molecular Ecology Resources, https://doi.org/10.1111/1755-0998.13139).

I look forward to reading your revised manuscript.

Reviewer 1 ·

Basic reporting

Throughout the text, there are several times when the correct plural word form or tense is not used. Check sentence structure and also double spaces.

Experimental design

The design is a little unclear and as I detail in the comments below, it needs to be clarified or reanalyzed so that the results are correctly interpreted. The use of dataset A, B, C is a good idea to help the reader understand but it appears pseudoreplicates have been used in the place of true replicates and the analysis may be void or not interpretable.

Validity of the findings

The premise of the article to investigate the invasive range of a relatively newly introduced toad compared to a native frog in the highly diverse and endemic region of Madagascar is a good one. The findings are currently a little unclear due to the analyses, yet the use of the statistical techniques is correct - ANOVA, UNIFRAC, PERMANOVA, NMDS, LEFSE, PICRUST and comparison to Bd database are all appropriate techniques for the data, however, some need to be reanalysed or interpreted so that findings can be interpreted correctly.

Additional comments

Abstract
Suggest adding common name of the toad in the abstract and again in the introduction at Lines 63-64
Introduction
Line 99 – don’t need to capitilise T in toad
Line 105 – forests of Eucalyptus spp.
Line 107 – is this the same as Parc Ivoloina?
Line 108 – I suggest that this sentence - D. melanostictus is known to have a high invasive potential - might be better as the start of a new paragraph. Perhaps there could also be a sentence as to why these toads are considered highly invasive.
Line 122 – include the frog common name
Methods
Line 133 – why 3-4 individuals? Below you state 4 individuals were used – needs to be clarified. And if they were sexed or not this should be mentioned.
Line 132-135 – you state 16 individuals for swabbing, yet in Table S2 there are far more than 16 in total – is this including multiple swabs from each individual? Also in Table S2, for DATASET A there are 9/9 at Site 1 for D. melanostictus. It is not clear to me what this table is referring to. The sample numbers are important when considering the downstream analysis, if sample replicates from the same individual are used in an ANOVA (or PERMANOVA) the result will not accurately represent the data because these are not true replicates, instead they are pseudo replicates with dependency on one another (i.e. from same individual if the case). From reading the methods section and adding the sampling numbers up, it is not clear to me how the analyses were performed or if they were performed appropriately. It is also important to check the homogeneity of variance after a significant p-value results as was found here.
Line 136-138 – what is the meaning/interpretation of this sentence During sampling at these sites, only the most generalist native amphibian species could be found (e.g. P. mascareniensis and Boophis tephraeomystax). ? This does not seem like information relevant to the methods and may do better in the discussion.
Lines 142-45 – it is interesting that site 3 appears to be more urban and yet this is where the invasive toad was not detected in 2014 although in the introduction it states they dominate urban areas. Suggest this could be clarified.
Figure 1 – suggest using shapes and colours in site identification and not just colours to suit people who may be colour blind.
Line 152 – were gut samples pooled from same sex or different sex and was this the same for both species?
Line 152-156 – this is a little confusing, were they stored in RNAlater because I understand this allows. storage at ‘room temp’. The final sentence refers to swabs but was this prior to RNALATER storage. Please clarify and be clear about the timeline of sampling and storage methods as multiple changes in temperature can impact DNA and downstream microbiome analyses.
Line 168 – what two replicates are being referred to here?
Line 173 – 174 – please provide a link as the raw sequences are not found under that Bioproject on NCBI.
Line 177 – low-quality reverse reads aren’t always the case and may still provide better quality when the reads are merged.
Line 194 – suggest you included the summaries on dataset groupings written further down before discussing the rarefaction.
Line 195 – this sentence doesn’t make sense, perhaps there is an extra word included?
Line 237 – is there a need for animal ethics approval here too?

Results
Need to clarify how the ANOVA was computed for diversity as mentioned above because based on the information it seems pseudo replicates were treated as replicates. There is also no mention of PERMANOVA in the methods or how this was conducted or how replication occurred and the measure of homogeneity of variances. There is no mention of the plots used, i.e. NMDS in the methods.
Line 281 – Thia to This
Line 285 – Better to refer to the species or its common name and not the genus since that is what you are referring to in this case.
Fig 3 – not sure why it is presented with multiple taxonomic levels in the LDA/LEFSE plot – better to show one taxonomic level or if this is the lowest level of classification of an OTU, show the identity of the OTU and make that clear so that it can also be compared between both species of amphibians.
Line 312 – were to where
Line 315 – was to were
As with Dataset A, it is unclear to me whether pseudoreplicates are used in the testing of sex differences. As you have no site 2 in one of your sexes, I suggest you test between male and female only using a one way design and remove replication by collapsing pseudoreplicates of repeat samples from an individual toad. Without complete knowledge of how the testing was conducted I can’t comment but based on the number of data points in the NMDS and the methods, these are pseudo replicates and not true replicates.
Line 321-323 – suggest rewording of this sentence, doesn’t make complete sense.
Line 324 – functional and Woodhams analysis need to be described.
For Dataset C as there is no replication, suggest you use the Monte Carlo p-value for PERMANOVA
Line 341 – needs rewording. Suggest: “Higher abundances of the classes Bacteroidia and Clostridia were found in ….”

Discussion
There are some more recent studies in this area that may be relevant to discuss related to Rhinella marina. The lack of observable significant differences (if that is the case once analyses are redone) could be due to lack of replication and should be mentioned in the discussion. It is important to identify when talking about skin or gut microbiome in the discussion. You are not able to say there is a trend towards a different gut microbiome – there is no data that supports this.

Reviewer 2 ·

Basic reporting

No comment

Experimental design

The primers’ (515F and 806R) sequences are different compared to the published ones (such as Caporaso et al. 2011) (the primers here seem reversed), which should be double checked.

Validity of the findings

No comment

Annotated reviews are not available for download in order to protect the identity of reviewers who chose to remain anonymous.

·

Basic reporting

This manuscript was generally well-written. There are a few instances of grammatical errors, which I have noted in the general comments. The figures are aesthetically pleasing and well-designed, though I have some comments on them below.

Experimental design

The methods and results sections of this manuscript seem more complicated than they need to be, mainly due to the separation of the data into three datasets analyzed with different rarefaction levels. It is unclear whether all these different rarefaction levels are necessary. If you run all of the analyses with just the lowest level of rarefaction that you use, will you get the same results? With just one large dataset, you could still separate out the two sample types (skin, gut), or even run the stats with all of the data and use posthoc tests to detect finer scale patterns.

There are some aspects of the methods that need more detail (see below).

Validity of the findings

no comment

Additional comments

This manuscript described skin and gut bacterial communities on a non-native toad from three sites in eastern Madagascar, comparing them with those on a native frog in the area. The paper is generally well-written and tries to put microbial patterns into the context of the toad invasion. Understanding the role of microbiomes in the spread of non-native hosts is important, and this paper provides an interesting comparison between a native and non-native frog. I have a couple larger comments on the manuscript and its organization, followed by more minor edits needed. Some of these comments are merely suggestions, which I think would help the flow and focus of your paper.

Hypothesis: Can you elaborate on why you hypothesized higher richness/diversity on the non-native toad? The references you cited in the discussion don’t support/lead to this hypothesis. It’s difficult to make predictions here based on previous research, since there hasn’t been much, but even if you added something to your hypothesis like, “because it would be comprised of a mix of microbes from the toad’s native and introduced habitats,” it would help the reader to follow your train of thought. It’s not clear how the factors you listed as causing this predicted pattern (representing distinct ecology from the other species investigated) tie into your hypothesis. Having different microhabitat preferences and different behaviors don’t inherently make toads have more diverse skin bacteria. Furthermore, it’s interesting that you analyzed the interaction between species and site in your permanova. If you hypothesized that the communities would differ between the species but not at all sites, or not in a similar way among the sites (because of the length of time the toads have been at the sites), that would also be interesting to note with the hypothesis.

Methods/Results: The methods and results sections of this manuscript seem more complicated than they need to be, mainly due to the separation of the data into three datasets analyzed with different rarefaction levels. It is unclear whether all these different rarefaction levels are necessary. If you run all of the analyses with just the lowest level of rarefaction that you use, will you get the same results? With just one large dataset, you could still separate out the two sample types (skin, gut), or even run the stats with all of the data and use posthoc tests to detect finer scale patterns.

If you simplify the larger dataset (use one instead of three), you could also subsequently simplify the results section. While patterns in skin and gut microbes should be addressed separately, there’s no clear reason why sex differences need their own section except due to your use of a separate dataset for sex analyses. For each type of analysis of the skin dataset, you could follow the results of species and site with whether or not the two sexes of toads differed. Condensing in this manner would dramatically help the flow of the results section. I also think that the results that are less discussed (LEfSe and KEGG pathways) could be given less space in the main manuscript.

Throughout the manuscript, there are places where providing some reasoning behind your methods and reminding the reader of that reasoning would go a long way in telling your story. For example, on line 204, you could add in something like, “We assessed alpha and beta diversity metrics to detect differences between the species, sites, and sexes.” The end of line 221 could read, “We used PICRUSt to gain a better understanding of the possible functions of the bacteria in our skin and gut samples.” Similar context and reasoning, by adding sentences or introductory phrases, would also be useful in the results to make the manuscript easier to follow.

The supplemental figures need an associated file of captions. It’s not clear what all the figure parts represent otherwise.

Figures: I think the manuscript would look cleaner if you only kept figures 1,2,5,and 6 in the main paper, and put the rest into the supplemental. For Figure 4, you could easily report the averages and standard deviations in the paper text.
Figures 2c, 5c, and 6c – instead of only looking at the top 10 taxa as 100%, it’s important to be able to visualize what proportion of the total the top 10 taxa comprised. Consider adding an “other” portion of the bar charts such that the entire community is 100% and the top 10 comprise just a portion of that total. In the text (Lines 268-274, 315-320, 339-347), you discuss the relative abundances of different taxa, but because the figure seems to re-scale the total community to only include the top 10 taxa, it’s not possible to know how the relative abundances differ between the species/sites, since the “other” category has been discarded. Also in these figures, I see that there’s a portion of phyla that are unidentified/unclassified. This is very rare if only the bacterial OTUs are kept. Make sure you’re only keeping “Bacteria” in your OTU filtering, and add that step to the methods.

Discussion: You did a good job of comparing your results with others published, and the discussion has a lot of detail. For the reader, it would be helpful to condense, or even remove, some of this information to make the discussion more focused. While explaining every result has its merits, focusing on your hypothesis is important. There are a few places where you could simplify without losing much content while gaining better flow. For instance, on line 400, you could elaborate on why you’d expect the toads to differ between the sites, and remove the sentence that goes from 400-402. The rest of this paragraph could also be simplified and condensed.
The question I kept asking myself while reading this ms was whether the fact that they’re new to the environment means they’re less likely to be selecting for any specific microbes on their skin, allowing them to harbor/keep a wider variety of the microbes they come across. On lines 423-424, you write that a richer microbiome could aid in host spread to new environments, but I wonder if the opposite could also be true.

Regarding the comparison with the Woodhams et al. 2015 anti-fungal dataset: In the methods, you should note what % similarity you used as a cut-off for your comparison against the Woodhams dataset. Did you re-run closed-reference OTU picking? For your results and discussion of these results, you can identify which of the taxa you detected were initially isolated from Madagascar, or toads elsewhere (since no toads from Madagascar are represented in that data set). In the discussion (~line 454), it’s interesting that the high prevalence of chytrid in India could be associated with the lower proportion of Bd-inhibiting bacteria. If they’re not selecting for inhibitory bacteria in the face of chytrid, do they have other forms of resistance/tolerance (more of a curiosity, not necessarily needed to add to your manuscript)?


Minor suggestions/comments:
Title – Including your results into a more descriptive title would go a long way in bringing in readers
Lines 55, 205 - semicolon should be a comma
Line 62 – ecosystem’ missing the final “s”
Line 88 – would be helpful to indicate that the insect in question is invasive
Lines 132-133 and 322-323 – There’s a discrepancy between the sample sizes (8 male and 8 female per site, but then no females from site 3). Please identify the number of samples analyzed somewhere other than the supplemental tables (at least a range of sample sizes per sex/site)
Line 139 and Fig1 caption – fix “Moore et al. (Moore et al. 2015)”
Lines 150+ – the first time you use “gut,” please explain what part of the gastro-intestinal tract you mean. Some authors use gut to mean the entire GI tract, others the entire intestine, and others just a portion of the intestines. Also, were there any further steps for the gut samples, such as opening them and rinsing out the gut contents? It would be important to note whether the gut microbes in your study are also representing whatever the animal has eaten.
Line 169 – when you pooled the samples “according to band size,” did you quantify the band brightness to do it?
Lines 198, 200 and 201 – either here or in the results, state what # of reads you rarefied to for datasets B and C
Lines 207-209 – Alpha diversity metrics often don’t fulfil the assumptions of parametric tests, especially with your lower samples sizes of dataset C. Consider replacing these with non-parametric tests.
Line 211 – cite the phyloseq package
Line 212 – what predictor variables did you include in your PERMANOVAs? From the results, it seems like you analyzed the interaction between species and site, but that’s not explicitly stated.
Lines 253-256, elsewhere – for non-significant p-values, report fewer decimal places to make it look less messy.
Line 251 – give the p-value for Shannon diversity ANOVA, not just <0.05, since you elsewhere write out significant p-values
Line 260, Figure 2b – it’s surprising that the weighted unifrac permanova detected differences between the species, because in the figure it looks like the two species greatly overlap considerably. Consider double checking this result
Line 281 – fix “Thia”
Line 284 – Figure 2b does not show this.
Lines 285-288 – Figure S2 doesn’t have a “b”. Was the wrong figure s2 uploaded?
Line 291 – You write that the frog only had OTUs from Proteobacteria, but Figure 2C clearly says otherwise. I think you mean that the differentially abundant OTUs were all in Proteobacteria. Clarify the wording in this paragraph.
Lines 311+ – Instead of writing that there were “no significant differences…except,” you could write that Shannon was the only alpha diversity metric that was significantly different between the sexes. With these sex analyses, it’s unclear why you would repeat the analyses that determine if site were significant, since you already analyzed that variable with dataset A.
Line 315 – change from “factors was” to “factors were”
Lines 365, 367, 369 – fix the citations
Line 374 – change “impoverish” to “impoverished”
Line 378 – this comma seems misplaced
Lines 399-404 – check your grammar in these sentences
Line 404 – add “in” after “investigated”
Line 406 – change “comparing” to “compared”
Line 420 – change “as” to “such as”
Line 460 – change “Similarly, to” to “Similar to”
Figure 2 – For people less familiar with frogs, consider adding species names somewhere on the figure itself.

---

## Round 0.2 · Minor Revisions

Thank you for your careful consideration of reviewers' comments. Your manuscript is nearly ready for publication. R3 has made a number of minor suggestions that would improve your manuscript. Please address these minor suggestions in your next revision.

Reviewer 1 ·

Basic reporting

No comment, the authors have successfully revised their manuscript.

Experimental design

No comment.

Validity of the findings

No comment

·

Basic reporting

This manuscript describes microbiomes in and on an invading toad species in Madagascar, further comparing the communities between the toads and a native frog species. The manuscript is well written, with sufficient literature cited, and I suggest some minor edits below.

Experimental design

The research question here was adequately explained, and besides the low sample size for gut microbiome comparisons, the research goals were generally well assessed.

Methods notes:
Line 145-146 – What is meant by “swabs were kept dry”? Were they air dried, or just placed in tubes without preservative? If not "dried", state how long they were stored at 4C. I have concerns that not freezing them sooner could have affected the communities. I am aware of a couple studies assessing effects of short-term storage at 4C but not longer term storage (e.g., Kim et al. 2017 Microbiome, Lauber et al. 2010 FEMS Micro Letters)

I’m not convinced that LEfSe is very meaningful with the small sample sizes of the gut samples. Because of the small sample size and pooled samples, it is important that not too much weight is put on the quantitative results from the gut microbiome data.

Validity of the findings

No comment

Additional comments

Minor comments:
It would be helpful to stick to either using the Latin or common names for your study species throughout the manuscript (preferably common name). There is a lot of jumping back and forth between the two, within and between sentences and paragraphs.

In the discussion, you should acknowledge that similarity to bacteria that can inhibit Bd does not necessarily mean that the bacteria on your toads actually have, or could have, that function. Your 97% similarity cut-off for identifying potentially Bd-inhibitory/enhancing bacteria could include a wide amount of diversity in those species.

Lines 32-33 – It’s not clear what “greater differences at lower taxonomic levels” means

Line 52 – Be careful about invasive species versus alien or non-native species. By definition, invasive species have negative effects on some aspect(s) of the ecosystem they’re invading.

Line 55 – ecosystem’ to ecosystem

Line 140 – If you want to re-introduce the main study species in the methods, input the full Latin name earlier (line 133-134), not here.

Line 148 – euthanized to euthanize

Line 150 – input sample sizes, or a range, for P. mascareniensis (per pooled sample)

Line 162 – we amplify to we amplified

Line 189 – OTUs to OTU

Line 201-202 – the note of “…, for a total of 12 analysed individuals of D. mel. And 12 individuals of P. mas)” is confusing, since you didn’t analyze 12 individuals, you analyzed pooled samples. Suggest removing this information.

Line 209 – effect of the sex  effect of sex

Line 219-220 – instead of “lack of replication”, maybe clarify with “low sample size and pooled samples do not allow for thorough statistical analysis…”

Line 252 – OUT picking to OTU picking

Line 256 – by proportions, do you mean proportions of OTUs or proportions of the total reads? This isn’t clear here and in the results. Also, clarify that these bacteria are “potentially” Bd-inhibitory/enhancing

Often in the results you refer to many panels of a figure, when you could refer to fewer at a time or in some cases, refer to those figures fewer times. For example, if line 269 refers to Fig 2a-d, don’t need to refer to them again at the end of the sentence. Another example, line 280 could refer just to Fig 2E, with line 281 referring to Fig 2F, or refer to them both once.

Line 286-287 – Which distance matrix are you referring to for the analysis of dispersion? Since both weighted and unweighted PERMANOVA were significant, should run this analysis with both distance matrices.

In lines 290-298, 340-344, and 371-380, you eye-ball differences between groups in relative abundance of different taxa, but then later provide the LEfSe analysis results. This is especially notable in the males vs females comparison, where you list differences, but then state that there were no differentially abundant taxa in either sex (line 346). I suggest you move the lefse results up to where you describe the communities, and you can specify, if need be, where taxon abundances appear different even though not significantly so.

Line 317 – taxa to taxon

Line 336 – missing a parenthesis

Line 337, 338 – consider replacing “using” (or line 338’s “using the”) with “assessing” or “comparing” or something similar

Line 354 – exhibiting to exhibited

Line 374 – double check the “10 abundant families”. Looks like 9, since “unclassified” isn’t a family, and may represent many families

Line 375 – Based on what you consider to be high abundance, it looks like P. mascareniensis has 7 families with fairly high abundance. Are you saying there are two families with higher relative abundance than in the toads, or that there were only two families with high abundance?

Line 385 – was to were

Line 405 – “on the gut microbiome the invasive species” to “on gut microbiomes of invasive species”

Line 407 – influence to influenced

Line 412, 540 – you didn’t really look at how the microbiomes “changed” across the toad’s expansion range, just looking at how they differ.

Line 413 – what makes a site “impoverished”? Not clear what you mean here.

Line 414 – host “identity” to host “species”. Identity could be read to mean individual frogs/toads

Line 415 – the “and at a lesser extent with sex” reads very out of place here. Missing a word maybe?

Lines 430-432 – Why do you think urban/town environments would impoverish the bacterial pool? Are there references to show this? The toads, which were found in more diverse, human-use areas, had greater bacterial richness than the frogs, found in the grass.

Lines 439-440 – It’s not clear what the second axis is in the “trend” of increase in alpha diversity across toad sites. The sites don’t seem to represent a gradient. Maybe better to say that the local populations differ?

Line 460 – assessed to assess

Line 498 – amphibian’ to amphibian

Lines 508-510 – Specify that you’ve switched topics to the gut microbiomes

Line 542 – change “more diverse than the one of the” to “more diverse than a”

Line 543 – toad to toad’s

In the taxa bar charts, consider having higher % total abundance cut-offs for inclusion. There are some colors in the legends that aren’t visible in the figures.

Including asterisks in figures to denote significant differences, or providing the statistical results in the figure captions, would be helpful.

---

## Round 0.3 · accepted · Accept

Thank you for addressing these final reviewer suggestions.